# Fruit Juice Spoilage by *Alicyclobacillus*: Detection and Control Methods—A Comprehensive Review

**DOI:** 10.3390/foods11050747

**Published:** 2022-03-03

**Authors:** Patra Sourri, Chrysoula C. Tassou, George-John E. Nychas, Efstathios Z. Panagou

**Affiliations:** 1Institute of Technology of Agricultural Products, Hellenic Agricultural Organization DIMITRA, Sofokli Venizelou 1, 14123 Lycovrissi, Greece; patrapsourri@gmail.com; 2Laboratory of Microbiology and Biotechnology of Foods, Department of Food Science and Human Nutrition, School of Food and Nutritional Sciences, Agricultural University of Athens, Iera Odos 75, 11855 Athens, Greece; gjn@aua.gr

**Keywords:** *Alicyclobacillus*, fruit juice, spoilage, detection, identification, control

## Abstract

Fruit juices have an important place in humans’ healthy diet. They are considered to be shelf stable products due to their low pH that prevents the growth of most bacteria. However thermo-acidophilic endospore forming bacteria of the genus *Alicyclobacillus* have the potential to cause spoilage of commercially pasteurized fruit juices. The flat sour type spoilage, with absence of gas production but presence of chemical spoilage compounds (mostly guaiacol) and the ability of *Alicyclobacillus* spores to survive after pasteurization and germinate under favorable conditions make them a major concern for the fruit juice industry worldwide. Their special characteristics and presence in the fruit juice industry has resulted in the development of many isolation and identification methods based on cell detection (plating methods, ELISA, flow cytometry), nucleic acid analysis (PCR, RAPD-PCR, ERIC-PCR, DGGE-PCR, RT-PCR, RFLP-PCR, IMS-PCR, qPCR, and 16S rRNA sequencing) and measurement of their metabolites (HPLC, GC, GC-MS, GC-O, GC-SPME, Electronic nose, and FTIR). Early detection is a big challenge that can reduce economic loss in the industry while the development of control methods targeting the inactivation of *Alicyclobacillus* is of paramount importance as well. This review includes a discussion of the various chemical (oxidants, natural compounds of microbial, animal and plant origin), physical (thermal pasteurization), and non-thermal (High Hydrostatic Pressure, High Pressure Homogenization, ultrasound, microwaves, UV-C light, irradiation, ohmic heating and Pulse Electric Field) treatments to control *Alicyclobacillus* growth in order to ensure the quality and the extended shelf life of fruit juices.

## 1. Introduction

Fruit juices are the most popular beverages, representing a significant market share within the food industry, and have an important role in human diet since their particular combination of physical and chemical characteristics render them natural and healthy [1]. They are low calorie foods rich in nutrients and bioactive compounds such as proteins, vitamins, carbohydrates, polyphenols, minerals, enzymes, fibers and antioxidants that can fit in today’s busy life style [2]. The full definition for fruit juice is “the fermentable but unfermented product obtained from the edible part of fruit which is sound and ripe, fresh or preserved by chilling or freezing of one or more kinds mixed together having the characteristic color, flavor and taste typical of the juice of the fruit from which it comes” [3]. Fruit juices can be classified according to their composition as fruit juice, fruit juice from concentrate, concentrated fruit juice, water extracted fruit juice, dehydrated/powdered fruit juice and fruit nectar. Depending on their dispersion system composition they are divided into clear, opalescent, cloudy and pulp enriched juices. According to the preservation method employed in order to prevent spoilage (microbial, chemical and enzymatic), while retaining their quality and nutritional value, they are classified as freshly squeezed, chilled, frozen, pasteurized and concentrated [4,5]. Today consumers prefer fruit juices as an easy way to cover the five servings of fruits and vegetables recommended by the World Health Organization for a healthy diet. The variety of different juice products on the market in combination with the use of new preservation technologies make them even more attractive. Furthermore, due to their low pH value, fruit juices do not favor the survival of pathogenic and spoilage microorganisms, making them safer and therefore more attractive to consumers [6,7].

## 2. Spoilage and Safety Aspects of Fruit Juices

In recent years many outbreaks concerning fruit juice contamination have been reported and the fruit juice industry has suffered financial damage [8,9]. Pathogenic and spoilage microorganisms are a challenge for the fruit juice manufacturers. The type of microorganisms present in fruit juice can originate from the fruits before harvest, therefore fallen fruits or fruits wounded from insects should be avoided. Other sources of microbial contamination could be the added water, flavorings or other chemicals, and finally process machinery and filling lines with deficient hygiene protocols. The relevant microorganisms considered as threats for commercial fruit juices are yeasts, molds and bacteria, while protozoa and viruses can also cause problems to a lesser extent [5,10].

Yeasts are the predominant spoilage microorganisms in fruit juices [11]. Their high acid tolerance and preference for anaerobic conditions, in combination with the sugar content and the refrigeration temperature during distribution and storage of the juice, favor spoilage incidents [12,13]. Contamination of fruit juices with yeasts results in carbon dioxide and alcohol production, increasing turbidity and flocculation, off-odors and changes in color [14]. It has been proved from previous researchers that representatives of *Saccharomyces*, *Candida*, *Zygosaccharomyces*, *Torulaspora*, *Rhodotorula*, *Hanseniaspora*, *Pichia* and *Trichosporon* genera are most frequently encountered in fruit juices [5,14,15]. The occurrence of contamination from yeasts in the fruit juice industry could be attributed to highly contaminated raw materials, failure in the pasteurization process and poor hygiene practices [16].

Moulds are microorganisms frequently encountered in fruit juices [11]. They are aerobic microorganisms that prefer low pH and high sugar content for growth [13]. Depending on their response to thermal treatment, moulds can be classified into heat resistant and heat sensitive [14]. The dominant heat resistant molds that have appeared in fruit juices over the years are *Aspergillus ochraceus*, *Aspergillus tamarii*, *Aspergillus flavus*, *Byssochlamys nivea*, *Byssochlamys fulva*, *Paecilomyces variotii*, *Neosartorya fischeri*, *Eupenicillium brefeldianum*, *Phialophora mustea*, *Talaromyces flavus*, *Talaromyces trachyspermus*, *Thermoascus aurantiacum*, *Penicillium notatum*, *Penicillium roquefortii* and *Cladosporium* spp. [17,18,19,20]. These moulds can produce gas, form colonies and floating mycelia on the surface, and change the odor of the juice [13,16]. Furthermore, they can cause disintegration to the fruit juice since they have the ability to produce disintegrative and pectinolytic enzymes [21], such as amylases, cellulases, pectinases and proteinases [5]. The most frequent heat sensitive molds belong to the genera of *Aspergillus*, *Penicillium*, *Mucor*, *Alternaria*, *Cladosporium*, and *Botrytis* [16,22]. Although these moulds can be eliminated with the pasteurization process [23], their presence indicates high contamination in the raw material or insufficient hygiene conditions during the manufacturing process [14]. Moulds are also associated with the production of mycotoxins, which is a serious safety issue for the fruit juice industry. They are secondary metabolites produced by fungi growing on food matrices. A mould has the ability to produce different kinds of mycotoxins and on the other hand one mycotoxin can be produced from different kinds of moulds [24]. The most dominant mycotoxins concerning the fruit juice industry are patulin and ochratoxin A. Patulin is mainly associated with apple juice and ochratoxin A with grape juice [25,26,27]. The maximum level for patulin in apple juice was established as 50 ppb [28], while the European regulation [29] recommends a maximum level of 25 ppb for solid apple products. The maximum concentration of ochratoxin A in grape juice and grape juice ingredients in other drinks has been defined as 2 ppb [30]. These mycotoxins could be a serious threat to human health worldwide due to their toxicity.

Bacteria are another group of microorganisms that has been associated with spoilage in fruit juices [7]. The acidic pH of most fruit juices favors the presence of lactic acid bacteria (LAB) and particularly the genera *Lactiplantibacillus* and *Leuconostoc* [7]. They produce off flavors similar to buttermilk and metabolic products such as lactic acid, formic acid, acetic acid, ethanol and carbon dioxide [16,31] that can change the juice flavor. Furthermore, their presence can cause haze and gas in the product [16]. Acetic acid bacteria are frequently found on many fruit surfaces and therefore associated with the spoilage of fruit juices, with *Acetobacter pasteurianus* and *Acetobacter aceti* being the predominant species [23]. These bacteria produce acetic acid from ethanol, sauerkraut and buttermilk off-flavors and can also cause browning of the juice [14]. Although both LAB and acetic acid bacteria are heat sensitive and can be destroyed with pasteurization [32], their presence indicates insufficient cleaning and sanitization of the equipment throughout the production line [14,33]. In order to avoid contamination, high standard hygiene protocols should be applied throughout processing [34]. Spore forming bacteria are a major issue for the fruit juice industry, since they can cause spoilage and cannot be controlled with standard pasteurization. The main problem from this group is due to *Alicyclobacillus* spp. and its predominant species *Alicyclobacillus acidoterrestris* [5]. Except from spoilage bacteria, pathogenic bacteria could be also considered as a threat to the fruit juice industry. Although fruit juices have been considered safe throughout the years, several foodborne outbreaks have been reported especially with unpasteurized fruit juices [35]. *Escherichia coli* O157:H7 [36,37,38,39], *Salmonella* [37,40,41,42,43] and *Staphylococcus aureus* [31] are considered to be implicated in many outbreaks of unpasteurized fruit juices including cider, apple juice and orange juice. Although *Listeria monocytogenes* has not been considered as a pathogen implicated directly in fruit juice outbreaks, it should be taken into consideration since it has the ability to survive throughout the production line of fruit juices [44]. In order to avoid the presence of pathogenic bacteria, the industry must retain high standard protocols of hygiene throughout the production line [17].

Protozoa are another threat for the fruit juice industry. The parasites of concern are *Cryptosporidium parvum* and *Cyclospora cayetanensis* that cause diarrhea [7], and the protozoan *Trypanosoma cruzi*, which causes Chagas disease affecting the autonomous nervous system in the esophagus, the heart and the colon [45]. Unpasteurized apple juice and cider have been associated with outbreaks of cryptosporidiosis [46,47], while in Brazil *Trypanosoma cruzi* has been involved in outbreaks associated with consumption of bacaba juice [48] and acai palm fruit juice [49].

Viruses can also contaminate fruit juices. Norovirus and Hepatitis A have been associated with outbreaks involving fruit juices such as orange juice [50,51]. The transmission of the viruses passes through contaminated fruit or water that has come in contact with feces [14]. The presence of protozoa and viruses can be intercepted by good agricultural and manufacturing practices and implementation of HACCP [52].

### 2.1. Alicyclobacillus spp. General Characteristics

In recent years, *Alicyclobacillus* has become the most serious threat of the juice industry. The isolation of this bacterium from various acid thermal environments was reported for the first time in the USA by Darland and Brock [53] and in Italy by De Rosa et al. [54]. Based on a previous published work in Japan [55], the characteristics of these bacteria were very similar to thermo-acidophilic microbes containing unusual *ω*-cycloexane fatty acids as a major component in their membranes. This microorganism was classified as a new species of the genus *Bacillus* and it was named *Bacillus acidocaldarius* [53]. In 1981, thermoacidophilic bacteria closely related to *Bacillus acidocaldarius* were isolated from neutral soils [56]. The first isolates from a non-thermal source of this species were reported by Cerny et al. [57] after a spoilage incident of pasteurized apple juice in Germany in 1982. These isolated strains were similar to those reported by Hippchen et al. [56] but differed from *Bacillus acidocaldarius* in the use of carbon sources. Thus Deinhard et al. [58] proposed to name the new species *Bacillus acidoterrestris*. Poralla and König [59] identified another *ω*-alicyclic fatty acid microorganism that contained mainly *ω*-cycloeptane and named this bacterium *Bacillus cycloheptanicus*. However, after sequence analysis on the 16S ribosomal RNA genes of the three species (*B. acidocaldarius*, *B. acidoterrestris* and *B. cycloheptanicus*), results indicated that they were very similar to each other but distinct from any other *Bacillus* species. Consequently, it was proposed that these three species should be reclassified and the new genus was named *Alicyclobacillus,* in favor of the *ω*-alicyclic fatty acids in their membranes [60].

Throughout the years more *Alicyclobacillus* species have been described (Table 1), but according to many researchers the predominant spoilage species is *A. acidoterrestris*. *Alicyclobacillus* species are Gram positive, except for *A. sendaiensis* [61], non-pathogenic, thermo-acidophilic rod-shaped endospore forming bacteria [62,63]. They have the ability to grow in a temperature range of 20–70 °C, with the optimum between 40–60 °C, and in a wide pH range (2.0–6.0), with an optimum between 3.5 and 4.5 [64]. Although all species are anaerobic and the presence of oxygen is expected to influence the growth of the microorganism, there is no agreement in the literature about the effect of oxygen on bacterial growth. Cerny et al. [65] reported that the presence or absence of the headspace in the container made no essential difference in the growth of *A. acidoterrestris* and no spoilage was observed in either case. On the contrary, Walker and Philips [66] demonstrated that containers of apple juice without headspace showed significantly lower growth levels in comparison to those containing headspace. Siegmund and Pöllinger-Zierler [67] also verified that the presence of limited oxygen decelerated *A. acidoterrestris* growth in apple juice without preventing high cell concentrations.

The presence of *ω*-alicyclic fatty acids in the membranes of *Alicyclobacillus* species is the dominant characteristic that distinguishes them from other spore forming bacteria. Researchers have claimed that *ω*-cycloexane and *ω*-cycloeptane rings in fatty acids contribute to the strong heat and acid resistance of *Alicyclobacillus* [68]. It has been also stated that the presence of cyclohexane rings in membranes increased the acyl chain density, resulting in a denser packing of the lipids in the membrane core, structural stabilization and lower fluidity and permeability of the membrane. This probably justifies the maintenance of the barrier function of the membrane, thus protecting the microorganisms in acidic and high temperature environments by forming a protective coating with strong hydrophobic bonds [60,68,69]. Another characteristic that may contribute to the resistance to extreme environments is the presence of hopanoids in the cells of most *Alicyclobacillus* strains [56,70]. The hopane glycolipids are structurally similar to cholesterol, and they affect the membrane lipid organization due to a decreased mobility of the acyl chain lipids. Furthermore, this action is more advantageous at low pH values [70].

The heat resistance of *Alicyclobacillus* endospores has been associated with several other factors including temperature, pH and water activity. Specifically, the temperature of thermal treatment exerts the greatest influence on the heat resistance of endospores, since the *D*-value decreases with increasing temperature. In addition, pH and Total Soluble Solids (TSS) also affect the heat resistance with a linear decrease in *D*-value with decreasing pH, and a linear increase in *D*-value when the content of TSS increases. Water activity also has an impact, since it has been shown that bacterial spores become more resistant as the values of a_w_ decrease [71]. Moreover, endospore resistance to heat can also be influenced by the presence of heat stable proteins and enzymes and the mineralization of dipicolinic acid (DPA) with divalent cations of calcium or manganese [68,72]. It needs to be noted that different strains even of the same species of *Alicyclobacillus* may have different *D*-values [73]. Furthermore, the cell number, the cell age, the sporulation temperature and the state of the endospore protoplast cortex can influence the heat resistance of the endospores [72,74,75].

Throughout the years, *Alicyclobacillus* species have been isolated from various environments, such as hot springs [60] and soils [76,77], as well as beverages, fruit concentrates and fruit juices [78,79,80,81,82]. The contamination of fruit juices by *Alicyclobacillus* species is most likely caused by soil, during harvest, as well as by fallen and unwashed or poorly washed fruits [68]. Employees can also transfer spores from the soil in the manufacturing facilities. Researchers have reported that water can also be a source of contamination [83,84,85,86,87] in the processing environment. Spoilage incidents of fruit juices by *Alicyclobacillus* species have increased considerably in the last years [78,79,81,88,89,90] including concentrated orange juice [6,91,92], apple juice [85,93], mango juice [94], passion fruit juice [95], pear juice [84,86], banana and watermelon juice [75], grapefruit and blueberry juice [96], and lemon juice [62].

The fact that spoilage due to the presence of *Alicyclobacillus* is difficult to detect makes this microorganism a serious problem for the fruit juice industry. Since there is no gas production or swelling of the container, contamination cannot be perceived until the consumer complains [68]. The evident sign of spoilage after consumption is an off-flavor described as medicinal, phenolic and antiseptic [69,97] associated mainly with the production of guaiacol (2-methoxyphenol), but also with the halophenols 2,6 dibromothenol and 2,6 dichlorophenol [63,98]. Guaiacol, which is the major metabolite associated with off-flavors in fruit juices, can be a product of microbial metabolism [68]. It is a spoilage compound produced during ferulic acid metabolism, from a non-oxidative decarboxylation of vanillic acid, catalyzed by vanillate decarboxylase [99,100].

### 2.2. Alicyclobacillus Acidoterrestris

Since its first association with spoilage in fruit juices in 1984 [57], *Alicyclobacillus acidoterrestris* has been considered as a challenge for the fruit juice industry worldwide [14]. It is the most important representative of the genus due to the number of reported spoilage incidents [63]. *A. acidoterrestris* has been isolated from a variety of juices and concentrates including apple, orange, lemon, mango, grapefruit, pear, tomato, white grape, pineapple, passion fruit, blueberry, pomegranate, cherry, strawberry, chokeberry, raspberry, watermelon, blackcurrant, kiwi and banana [82,92,95,118,119,120,121,122,123]. It is a spore forming bacterium that can survive thermal treatment during pasteurization, grow at low pH, germinate, and spoil the juice [124]. Therefore, it has been proposed as a target microorganism to control the effectiveness of the pasteurization process in acid fruit juices. The maximum accepted concentration of *A. acidoterrestris* spores as defined by the fruit juice industry is 100 CFU/mL of raw material [125]. *A. acidoterrestris* is an aerobic, Gram-positive, rod-shaped endospore-forming spoilage microorganism [62,63]. It can grow in a wide pH range (2.0–7.0) with the optimum between 3.5 and 4.0, and in a temperature range of 25–60 °C with the optimum between 40 and 45 °C [126,127,128]. *A. acidoterrestris* spores are very heat resistant and, depending on the conditions of thermal treatment and bacterial strain, D_90__°C_ ranges between 5.95 and 23 min [129] and D_95__°C_ between 0.06 and 8.55 min [130]. The main characteristic of *A. acidoterrestris* strains that make them so tolerant to heat is the presence of *ω*-cycloexane fatty acids in their membranes [60,87].

*A. acidoterrestris* spores have a slow growth cycle of ca. 5 days [62]. Spoilage is not visible during storage or retail since there is no gas production and swelling of the juice container (flat-sour type spoilage). Only after consumption, flavors described as “smoky”, “antiseptic” or “disinfectant” and possible increased turbidity and sediment formation can lead to the conclusion of spoilage of the juice [87,131,132]. The predominant spoilage compound responsible for this is guaiacol [131]. Although the contamination pathway with guaiacol from *A. acidoterrestris* has not been clearly elucidated, the most accepted assumption is that it is produced during ferulic acid metabolism [69,133]. Microorganisms usually decarboxylate ferulic acid to 4-vinylguaiacol [134] causing a “rotten” flavor especially in orange juice [135]. However, it can also be directly metabolized to vanillin [136] or vanillic acid [137]. *A. acidoterrestris* is capable of producing guaiacol from vanillin [138] and vanillic acid [139]. Although tyrosin and lignin have been suggested as precursors for the production of guaiacol from *A. acidoterrestris*, the pathway has not been studied extensively [69,133]. It has been proved that when the concentration of *A. acidoterrestris* cells ranges between 10^5^ and 10^6^ CFU/mL it produces enough guaiacol to spoil the juice [75,131]. Considering the substantial economic losses in the fruit juice industry due to the growth of *A*. *acidoterrestris* spores, the factors that induce spoilage should be seriously taken into account, specifically since spoilage is not apparent before consumption. The cell concentration of the microorganism, the heat shock treatment, the incubation temperature, the oxygen availability and the growth medium are among the factors influencing spoilage [68]. It must be noted that spore germination and growth is inhibited under 20 °C [140] and even at low oxygen concentration, contamination cannot be completely suppressed [67]. Furthermore, the growth behavior of *A. acidoterrestris* strains depends on the type of the juice and the isolation source of the strain [76]. However, the presence of *A. acidoterrestris* in juice is not a threat for human health, since neither the microorganism nor its metabolites have ever been associated with illness from the consumption of contaminated juice [141]. Although *A. acidoterrestris* is a non-pathogenic bacterium, spoilage incidents are a major concern for the fruit juice industry mostly because of the difficult detection of the bacterium, due to the absence of visible deterioration of the containers and associated spoilage.

## 3. Isolation and Identification of *Alicyclobacillus* spp.

Since spoilage from *Alicyclobacillus* has become a major issue for the fruit juice industry resulting in high economic losses, the need for developing rapid, accurate and sensitive methods for the early detection of the bacterium is of paramount importance. Initially researchers used mainly direct plating and spoilage detection methods, but nowadays detection methods based on instrumental analysis, immunodetection and molecular analysis are becoming more popular. Detection can be separated into three strategies, namely (a) targeting the cell/spore detection, (b) nucleic acid analyses, and (c) metabolites measurement [142]. An overview of the detection and identification methods for *Alicyclobacillus* is displayed in Table 2.

Plating methods are simple and reliable but cannot detect low populations of the bacterium. Since research findings indicate that *A. acidoterrestris* spores do not grow on acidified agar such as Brain Heart Infusion agar, Nutrient agar, Standard Plate Count agar, Tryptone Soy agar and Veal Infusion agar, new media have been developed in order to isolate and successfully enumerate these spores [69,171]. Thus, the media that favor the growth of *Alicyclobacillus* after being acidified to pH 3.5–5.6 by HCl, H_2_SO_4_ and malic acid after autoclaving [68,75] are *Bacillus acidocaldarius* medium (BAM) and *Bacillus acidoterrestris* thermophilic (BAT) agar [58,92,187,188], Yeast Starch Glucose Agar (YSG) [78,111,189], Orange Serum Agar (OSA) [190], K agar [85], Potato Dextrose Agar (PDA) [91,96,118] and SK agar [191]. It has also been suggested that spread plating is more effective than pour plating for bacterial growth [171,188,192], but Yokota et al. [64] reported the opposite on YSG agar. Although several traditional microbiological methods have been employed for the detection of *Alicyclobacillus* strains, the IFU Method No 12 developed by the Working Group on Microbiology of the International Federation of Fruit Juice Producers (IFU) is considered to be the most effective [193]. This method can distinguish spoilage and non-spoilage species but it is time-consuming. The membrane filtration technique is another method used in combination with the plating method, mainly when high populations of the bacterium must be detected [189]. IFU Method No 12 recommends the use of 0.45 μm filters, while the European Fruit Juice Association (AIJN) recommends 0.2 μm filters. Although this technique is more sensitive and has a lower detection limit [194], it cannot be used for all products [195]. Several isolation methods including the IFU Method No 12, apply heat shock treatment to the endospores of the microorganism in order to destroy the existing vegetative cells and induce the germination of predominant spores [64]. Many heat treatment schemes have been suggested but the predominant include thermal treatment at 80 °C for 10 min (recommended by the IFU) and at 70 °C for 20 min (recommended by the JFJA) (Japan Fruit Juice Association) [69,75,133]. Although these methods have been widely employed for routine analysis by the industry due to their low cost, they are time consuming and demanding, thus novel rapid detection and identification techniques are needed.

Enzyme-linked immunosorbent assay (ELISA) is a biochemistry assay that has been applied for the detection of *Alicyclobacillus* mostly in apple juice, using a specific polyclonal anti-*Alicyclobacillus* antibody [143,144]. This method reduces the detection time to 6–7 h but detects populations higher than 10^5^ CFU/mL. The above procedure has been improved by adding immunomagnetic separation that has shortened the time of analysis to 3 h and the detection limit to 10^3^ CFU/mL [196]. Even though ELISA is rapid and reliable, it has high cost of analysis and legal limitations of animal use for the antibody production [159].

Flow cytometry is another cell detection method based on laser light that scatters samples in order to obtain cell size and corresponding light patterns of DNA density. This method detects cell concentration higher than 10^3^ CFU/mL for *Alicyclobacillus* strains in fruit juice concentrates within 10 h [141]. Although the detection is achieved within limited time, flow cytometry can only be used for fluid samples [133].

Over the years methods based on Polymerase Chain Reaction (PCR) have been widely employed in research for the rapid identification of microorganisms and were also successfully applied to *Alicyclobacillus*. Reverse transcription polymerase chain reaction (RT-PCR) was first used by Yamazaki et al. [197]. Based on *shc* (squalene-hopene cyclase) gene, a key enzyme in the biosynthesis of hopaniods, researchers managed to detect *A. acidoterrestris* and *A. acidocaldarius* with a detection level of 1–2 CFU/mL after 15 h of enrichment. In 2004, Luo et al. [154] developed a Taqman^®^ RT-PCR method also based on *sch* gene with a detection level less than 100 CFU/mL within 3–5 h for both species. A Taqman^®^ PCR targeting the 16S rRNA gene was able to detect more species of *Alicyclobacillus* within 5 h and with a detection limit lower than 100 CFU/mL [160]. Random Amplification of Polymorphic DNA (RAPD) PCR has also been selected as a rapid method in order to distinguish *Alicyclobacillus* strains [77,82,95,151,152,198,199]. The selected primer and the lysed DNA of the microorganism are mixed with *Taq* polymerase and after PCR and electrophoresis the bands that appear are further analyzed [200]. Yamazaki et al. [198] reported the identification of *A. acidoterrestris* from acidic juice by applying RAPD PCR within 6 h. Restriction fragment length polymorphism (RFLP) PCR is another rapid and low-cost method that differentiates homologous DNA sequences, which are detected by different length fragments after DNA digestion and are then cut by restriction endonuclease. After gel electrophoresis, a unique fingerprint is received. Analysis of 16 S rRNA RFLP has been used for the characterization of *Alicyclobacillus* strains from concentrated apple juice [85] and orange juice [157].

16 S rRNA sequence analysis has been used widely for identification, because this gene deviates among the closely related *Alicyclobacillus* bacterial species. Furthermore, the 5′-end hyper-variable region of the gene varies among *Alicyclobacillus* species and makes it sufficient for discrimination among species [201]. Moreover, the immunomagnetic separation method (IMS), which is based on magnetic beads that capture the microorganism cells improved the sensitivity of PCR and RT-PCR when cooperating with the 16 S rRNA gene for the detection of *Alicyclobacillus* [164,165,196].

Denaturing gradient gel electrophoresis (DGGE) has been also proved to be effective not only for the detection of *Alicyclobacillus* but also for the distinction of guaiacol producing and non-producing species by adding an *Alicyclobacillus* DNA sequence ladder mix on the DGGE gel [80]. Vermicon Identification Technology (VIT), which is based on fluorescent labelled probes, has been shown to have a detection limit of 1 CFU/mL within 3 h of isolation and can be applied directly to fruit juice concentrate. This method is also capable of differentiating *Alicyclobacillus acidoterrestris* from other *Alicyclobacillus* species since they glow in different colors [202].

Indirect detection of *Alicyclobacillus* spoilage can be determined by measuring the metabolites, mainly guaiacol. The determination can be accomplished using sensory, analytical, or chemical methods.

Sensory methods are mostly used for screening the sample for the presence or absence of spoilage compounds. A trained panel is usually asked to describe the taste, aroma, sourness, color, and finally the acceptability of the sample when compared to a control (unspoiled) sample [174]. The published studies concerning the detection of guaiacol by a sensory panel showed that the detection is highly dependent on the sensitivity of the panelists and on the sample matrix, due to the variation of the components in fruit juices [67,171,174,175,180]. Although analytical methods are considered to be more sensitive to the detection limit of guaiacol, other researchers reported that sensory analysis presented greater sensitivity [171,203].

Analytical methods are used for both qualitative and quantitative detection and the most frequently used are chromatographic analysis such as Gas-Chromatography (GC) and High-Performance Liquid Chromatography (HPLC). These methods include three steps: extraction, separation and identification [68,69]. After collecting an adequate quantity of the sample, heat desorption or solvent extraction follows for GC or HPLC, respectively. The separation of the compounds with the use of specific columns depends on their molecular weight, solubility, ion exchange capacity and polarity, which emerge at different retention times. With the use of standards, the method can detect the chemical compound and its quantity [68]. The GC-MS (mass spectrometry) is widely employed due to the sensitivity of the detector [131,171,180] together with GC-O (olfactometry) [131]. Solid Phase Microextraction (SPME) has also been successfully combined with GC for the determination of volatile compounds [175,176,180,181]. Apart from GC, the use of HPLC for the detection of *Alicyclobacillus* spoilage has also been reported [138,167]. Although the former mentioned analytical techniques have been shown to be accurate, they are expensive, time consuming, require skilled personnel for operation and analysis of the results and cannot be adapted easily in the production line [133,142].

Electronic nose (EN) is an artificial sensing system, based on a chemical sensor array of semi selective gas sensors combined with pattern recognition algorithms. With the proper data analysis tools, EN could result in the early detection of *Alicyclobacillus* contamination. Gobbi et al. [172] detected *Alicyclobacillus* spp. in peach, orange and apple juice after 24 h from inoculation, while Cagnasso et al. [179] identified spoilage from *A. acidoterrestris* in orange and pear juice at the same time period. Concina et al. [162] identified the contamination of *Alicyclobacillus* spp. in commercial flavored drinks at the early stage of growth and Huang et al. [182] reported that EN could perceive a contaminated apple juice beverage after 4 h when coupled to linear discriminant analysis. It is a promising method because it is simple, quick, reliable and of low cost, and can be easily used in the production line [181].

Another rapid method that is widely used for the detection and identification of bacteria is Fourier Transform Infrared Spectroscopy (FTIR). FTIR is based on measuring distinct biochemical characteristics of the cytoplasm and the cell wall components, presenting them as different spectral features at 400–4000 cm^−1^. The detection limit of this method is 10^3^–10^4^ CFU/mL and it can distinguish different species of *Alicyclobacillus* and classify them as guaiacol and non-guaiacol producing strains [184,185]. Some drawbacks of this method include the cost of the equipment and an essential extension with comprehensive spectral reference database in order to limit detection time for unclassified *Alicyclobacillus* strains [133,142].

The chemical method that has been broadly used for the detection of guaiacol is Peroxidase enzyme colorimetric assay (PECA), which can detect and quantify the presence of guaiacol. PECA is based on the oxidation of guaiacol by peroxidase enzymes in the presence of H_2_O_2_ with the formation of a brown compound, identified as 3,3′-dimethoxy-4,4′-biphenoquinone [204], which can be measured by spectrophotometry at 420 nm [138,167] or 470 nm [204,205,206]. Τhe guaiacol detection kits that are available in the market are based on this method and besides detecting guaiacol they can also quantify it by using standard concentration curves. Although this method is simple, less time consuming and of low cost, it is imprecise and most frequently used only for the detection of the presence of guaiacol and not for quantification.

## 4. Control of *Alicyclobacillus* spp.

The spoilage of fruit juices from *Alicyclobacillus* spp. has been shown to start from the beginning of the supply chain, since contaminated fruits at harvest can intrude into the production line and cause problems that will appear only after consumption. This observation necessitates the implementation of highly effective measurements in order to avoid spoilage from the beginning and therefore economic loss for the fruit juice industry. Good Manufacturing Practices and systematic use of Hazard Analysis and Critical Control Points (HACCP) rules throughout the whole supply chain can control contamination from *Alicyclobacillus*. In order to ensure fruit juices with high safety and extended shelf life, chemical, physical, and combined methods have been developed.

### 4.1. Chemical Treatments

The first step to control the contamination from this microorganism is to avoid harvesting fallen fruit or at least wash the surface of the fruit properly with the use of disinfectants. The oxidants that are usually diluted in water are sodium chlorite (NaClO_2_), chlorus acid (HClO_2_) and chlorine dioxide (ClO_2_) [175]. Since 1998, the use of ClO_2_ has been allowed by the Food and Drug Administration [207] as an antimicrobial chemical and therefore it is widely used for the disinfection of fruit, containers, and processing equipment. The effectiveness of this sanitizer on the inactivation of *Alicyclobacillus* spores is possibly due to the injury of the inner membrane of the spore resulting in germination and outgrowth [208]. Bevilacqua et al. [209] also implied that this oxidizing compound aimed to damage the inner membrane of *A. acidoterrestris* spores. These disinfectants can also be used as preservatives in the fruit juice processing line [133].

Ozone (O_3_) is another oxidant also recognized as safe from the FDA that can be used in fruit juices. It also has the potential to eliminate *Alicyclobacillus* spores, since it has been shown that as the concentration and the treatment time of O_3_ increases, the inactivation of *A. acidoterrestris* also increases [210].

The growth of *Alicyclobacillus* can also be controlled with the use of some organic acids. The effectiveness of acids on bacterial cells, but not on spores, in ascending order was benzoic, butyric-caprylic, acetic, citric-malic-lactic, and tartaric acids [211]. Chemical preservatives like sodium benzoate and potassium sorbate have been allowed to be added in beverages with a limit of 1500 mg/L [212]. It has been reported that they can control *A. acidoterrestris* growth [103] with the need for higher concentrations for vegetative cells than for spores [213].

The increasing demand of consumers for natural additives in food products has led to the use of natural compounds in fruit juices for the control of *Alicyclobacillus*. Natural antimicrobials of microbial origin, animal origin and plant origin have been successfully used for the inhibition of the microorganism [133].

#### 4.1.1. Natural Compounds of Microbial Origin

Bacteriocins are antimicrobial peptides or proteins that are produced from various bacteria, which present antimicrobial activity against closely related species [214]. Nisin is a non-toxic polypeptide used in many countries as a safe food preservative [215,216]. It is obtained from *Lactococcus lactis* subsp. *lactis* and has a significant effect especially on the spores of *A. acidoterrestris* [89,129,217,218,219,220]. Nisin is the only bacteriocin used for the control of *A. acidoterrestris* in the fruit juice industry at present [214], added either directly in the juice [129,217] or integrated into the biodegradable polylactic and polymer film of the container [221,222]. Studies have revealed more bacteriocins to be effective against *Alicyclobacillus* including enterocin AS-48 produced from *Enterococcus faecalis* [223], bificin C6165 from *Bifidobacterium animalis* subsp. *animalis* [224], biovicin HC5 purified from *Streptococcus bovis* [225], warnericin RB4 from *Staphylococcus warneri* [226], paracin C from *Lactiplantibacillus paracasei* [227,228] and cyclin A from *Lactiplantibacillus plantarum* [229]. Although all of them have high potential in inhibiting *Alicyclobacillus* strains they have limited application in the industry due to the high cost of extraction and purification [133].

#### 4.1.2. Natural Compounds of Animal Origin

Lysozyme is an enzyme present in various biological tissues and fluids like tears, saliva, eggs and milk, often used to inhibit Gram positive bacteria and especially thermophilic spore forming bacteria at a concentration of 20 μg/mL [230,231]. Lysozyme is also considered as a safe preservative [232] and has been applied directly in fruit juices [233] or through the polymeric matrix film in packaging in order to control *Alicyclobacillus* [234,235]. The efficiency of lysozyme depends on the concentration, the strain of the bacterium and the external conditions applied [233,236,237,238].

Chitosan, the only basic polysaccharide in nature is a derivative of chitin, extracted from the shell of shrimps, crabs and crawfishes; it has the ability to control bacteria, yeasts, and molds [239,240]. When combined with thermal processing it can inhibit *A. accidoterrestris* spores from germinating at a concentration level of 1.4 g/L [241].

#### 4.1.3. Natural Compounds of Plant Origin

Essential oils (EOs) are aromatic liquids obtained by extraction, distillation, fermentation or enfleurage from plant materials, mostly herbs and spices that are used in the fruit juice industry as food flavorings [133,242,243]. The antimicrobial activity of cinnamaldehyde, eugenol and carvacrol has been reported to be efficient against *A. accidoterrestris* spores [90,244]. Lemon essential oil and extracts of *Eycalyptus maculata* also controlled the germination of *A. accidoterrestris* spores as reported by Maldonado et al. [245] and Takahashi et al. [244]. *Fatty acids and esters* have also been reported to have antibacterial activity against *A. accidoterrestris* spores. Manolaurin, which is recognized as a safe compound from the FDA, was effective against the vegetative cells of the microorganism [246]. Sucrose palmitate, sucrose stearates and sucrose laurates have also been reported as efficient antimicrobials against *Alicyclobacillus* spores [247]. Other plant extracts have been reported to have antimicrobial effectiveness against *Alicyclobacillus*. Saponin that was extracted from *Sapindus saponaria* fruits inhibited *A. acidoterrestris* spore germination, but affected the sensory quality and resulted in foam production [248]. Papain and bromelain enzymes extracted from *Carica papaya* and *Ananas comosus,* respectively, also controlled *A. acidoterrestris* spore germination [249], whereas two formulations of *Rosmarinus officinalis* were effective against *A. acidoterrestris*, *A. hesperidum* and *A. cycloheptanicus* vegetative cells [250]. An overview of chemical treatments applied for the inactivation of *Alicyclobacillus* species is presented in Table 3.

### 4.2. Physical Treatments

Thermal pasteurization is a heat treatment successfully employed by the fruit juice industry in order to extend the shelf life of processed fruit juices. It manages to inactivate heat sensitive microorganisms and enzymes that can degrade the quality without influencing the sensory characteristics of the fruit juice. The conventional heat treatment (88–96 °C for 2 min) however is not sufficient against *Alicyclobacillus*, since its spores can survive pasteurization treatments and germinate under favorable conditions during storage [68,71]. Provided that refrigerated temperatures are ensured throughout the whole supply chain, *Alicyclobacillus* spores would not grow since germination is inhibited at temperatures below 20 °C. Distribution though does not always take place under refrigerated conditions, due to high cost, therefore spores may induce spoilage during warmer months [68]. In order to control the presence of *Alicyclobacillus*, the temperature of thermal treatment should be increased, but this would result in quality (vitamin and nutrient loss) and sensory (nonenzymatic browning and flavor compounds) deterioration [251,252,253]. Consequently, the development of nonthermal methods is necessary to retain both safety and quality attributes of fruit juices.

### 4.3. Nonthermal Treatments

High hydrostatic pressure (HHP) is based on two principles, namely the isostatic and the *Le Chatelier*. The first one secures the homogeneous and instant distribution of the pressure applied equally in all directions of the sample and the second one states that any occurrence of chemical or biochemical reaction, molecular configuration or phase transition that can lead to volume reduction is improved by pressure [254,255]. The food industry uses HHP with pressure ranging from 100 to 800 MPa, duration from milliseconds to more than 20 min, and treatment temperatures from 0 to 90 °C. The main mechanism of HHP that causes the inactivation of *Alicyclobacillus* is the damage to the noncovalent bonds that are present in lipids, proteins, nucleic acids and polysaccharides. In this way, HHP affects the cell membrane constituents such as proteins, enzymes and ribosomes and therefore damages the genetic material of the microorganism, since it causes denaturation of cell components resulting in the injury and death of the microorganism [256]. HHP has been used extensively for the inactivation of *Alicyclobacillus* because when combined with heat treatment it is very effective against spores that are very resistant to inactivation. HHP can cause the germination of spores, which are less resistant to dormant spores that will be subsequently killed with the simultaneous heat treatment [257,258]. The germination of spores at lower pressures (50 to 300 MPa) proceeds via activation of nutrient receptors (d-sugars, l-amino and purine nucleosides), while at higher pressures (400–800 MPa) germination is triggered by the direct release of Ca-DPA (dipicolinic acid) [259,260]. It has been proved that cycle pressure treatments can enhance the inactivation of the spores for an equivalent duration of a single pressure application. Treatment with the low pressure exposure results in spore germination and the higher pressure inactivates spores and vegetative cells [261]. Relevant research concerning the application of HHP for the inactivation of *Alicyclobacillus* is summarized in Table 4. Although HHP is very effective, some dormant spores have the ability to remain immutable, a fact that must be taken into serious consideration by the fruit juice industry when applying HHP treatments [262]. Juices treated with HHP often exhibit superior quality compared to those treated with thermal processing since HHP has minimum impact on color, flavor and taste while retaining nutrients, vitamins, amino acids and functional properties [263]. Taking into account consumers’ demands for minimally processed products, HPP has many industrial applications and therefore it has been the most popular non-thermal treatment since the late 1980s.

High pressure homogenization (HPH) (150–200 MPa) or Ultra High pressure homogenization (350–400 MPa) is a food processing technology that is based on the principles of conventional homogenization with higher pressures [277] and can be applied only to fluid products. The shear stress distribution across the treated product is responsible for the changes occurring in microorganisms resulting in inactivation [5]. Bevilacqua et al. [126] reported reduction of *A. acidoterrestris* population with cells being more sensitive than spores when applying HPH (500, 800, 1100, 1400 and 1700 bar) for 2 ms to three different strains inoculated in malt extract broth. The susceptibility was proved to be strain dependent. This treatment has limited industrial applicability because of the need for refrigeration in order to guarantee the safety of final products.

Ultrasound or ultrasonic waves are electromagnetic waves with frequency above 20 kHz that can create cavitation in the cell wall of the microorganism and thus destroy it [278]. *A. acidoterrestris* vegetative cells in apple juice were inactivated with ultrasonic treatment that seemed to be more effective as the power level and the processing time increased [279]. Wang et al. [280] also reported that ultrasonic waves inhibited *Alicyclobacillus* vegetative cells and that the effectiveness of the method depended on the matrix, the strain, the power level, and the exposure time. Ultrasound treatment has been proved to be more effective when combined with other processes, in particular high pressure and heat. Although this treatment is considered to improve the quality of many products including fruit juices [281], it affects the sensory characteristics of the fruit juice and thus may not meet consumer’s demand [279].

Microwaves are also electromagnetic waves that have the ability to change the cell membrane permeability, break the hydrogen bonds of RNA and DNA and thus inhibit the cell growth [133]. Microwave sterilization has been employed as a nonthermal treatment since it heats the product faster without influencing the texture and the taste and does not lead to cell cortex swelling like conventional sterilization [133,282].

Ultraviolet (UV-C) light is a form of electromagnetic radiation ranging from 200 to 280 nm [283] that has the ability to damage the DNA of the microorganism and therefore eliminate it [284]. The treatment has been proved effective against *A. acidoterrestris* spores in grape and apple juice [285,286]. The low energy consumption and the absence of toxic byproducts in the final product makes UV-C light a promising control treatment and for this reason FDA approved its use in order to clear fruit juices (FDA, 2000).

Irradiation treatment uses gamma rays, electrons, or X rays [253,287] targeting the chromosome in order to split the double helix of DNA and thus damage the cell of the microorganism [253]. The use of gamma rays and electrons were reported to be effective against *A. acidoterrestris* spores in citrus juice in combination with heat treatment (85–95 °C) [288]. In addition, Lee et al. [289] reported the inactivation of *A. acidoterrestris* spores in apple and orange juice with the use of gamma rays. Irradiation has limited applicability in the fruit juice industry today due to its association with radioactivity that is unacceptable from the consumers’ point of view [277].

Ohmic heating uses an electrical current to generate heat instantly inside the food in order to kill microorganisms [73]. *A. acidoterrestris* vegetative cells were inactivated in apple juice with the use of an ohmic heating system. When the temperature was above 70 °C the death rate was close to 100% [290]. Moreover, the inactivation of *A. acidoterrestris* spores was reported to be higher with the use of ohmic heating than with conventional heating in orange and apple juice [291,292]. However, additional studies should be undertaken to verify the effectiveness of this method in the fruit juice industry.

Pulsed electric field generates pulse waves that have enough intensity to cause cell membrane damage that leads to cell destruction [293]. Uemura et al. [294] reported the reduction in the population of *A. acidoterrestris* in orange juice in a very short time (0.9 s) at 125 °C without influencing the nutritional quality of the juices. This technology has the ability to improve the microbiological quality and preserve the physicochemical and nutritional attributes of juices, but concerning *Alicyclobacillus* spores, more studies, including sensory assessment, should be performed in order to elucidate the effects of temperature assistance on the organoleptic traits of fruit juices.

## 5. Conclusions

Fruit juices have gained popularity due to their health benefits resulting in the expansion of the global juice market. Therefore spoilage incidents can cause significant financial losses to the industry. *Alicyclobacillus* and *Alicyclobacillus acidoterrestris* in particular are thermo-acidophilic spore forming bacteria responsible for spoilage that cannot be detected until consumption of the juice, making them a major hazard for the fruit juice industry. The quality of the raw material and the hygiene processing conditions should be taken under consideration to avoid spoilage. Subsequently, various control and prevention methods have been established to inactivate *Alicyclobacillus* spores and preserve the quality and the shelf life of the fruit juice. The early detection of spoilage using rapid methods is also a requirement of the industry. Consequently, future studies should focus on the improvement of the existing techniques and the development of new methods to ensure the rapid and early detection of *Alicyclobacillus* and preserve the quality of fruit juices.

## Figures and Tables

**Table 1 foods-11-00747-t001:** *Alicyclobacillus* species isolated from various sources.

*Alicyclobacillus* Species	Source	Reference
*A. acidiphilus*	Acidic beverage	[78]
*A. acidocaldarius*	Thermal acid waters	[53,55,60]
*A. acidocaldarius subsp. cidocaldarius*	Fruit juice or soft drink	[101]
*A. acidocaldarius subsp. rittmannii*	Geothermal soil of Mount Rittmann, Antarctica	[102]
*A. acidoterrestris*	Soil/apple juice	[56,58,60,87]
*A. aeris*	Copper mine, China	[103]
*A. cellulocilyticus*	Steamed Japanese cedar chips	[104]
*A. consociatus*	Human clinical specimen	[105]
*A. contaminans*	Soil, Fuji city Japan	[76]
*A. cycloheptanicus*	Soil	[58,59,60]
*A. dauci*	Mixed vegetable/fruit juices	[106]
*A. disulffidooxidans*	Water sludge, Canada	[107,108]
*A. fastidiosus*	Apple juice	[76]
*A. ferrooxydans*	Solfataric soil	[109]
*A. fodiniaquatilis*	Acid mine water, China	[110]
*A. herbarius*	Hibiscus herbal tea	[111]
*A. hesperidum*	Solfataric soil	[112]
*A. kakegawensis*	Soil, Japan	[76]
*A. macrosporangiidus*	Soil, Japan	[76]
*A. montanus*	Hot spring	[113]
*A. pohliae*	Geothermal soil, Antarctica	[114]
*A. pomorum*	Mixed fruit juice	[115]
*A. sacchari*	Liquid sugar	[76]
*A. sendaiensis*	Soil, Japan	[61]
*A. shizuokensis*	Soil in crop fields, Japan	[76]
*A. tengchongensis*	Soil in hot spring, China	[116]
*A. tolerans*	Oxidizable lead-zink ores	[108]
*A. vulcanalis*	Hot spring, United States	[117]

**Table 2 foods-11-00747-t002:** Detection methods of *Alicyclobacillus* species.

Detection Method	Isolation Source/Medium	Reference
Cell/Spore-Based Methods		
ELISA	Apple juice	[143,144,145,146]
	Apple juice concentrate	[147,148]
	Orange, clear apple, unfiltered apple, pear, tomato, pink grapefruit, and white grape	[149]
Flow Cytometry	Apple juice concentrate	[141]
**Molecular Methods**		
PCR	Isotonic water, lemonade, fruit juice blend, fruit carrot juice blend	[81]
	Orchard soil	[77]
	Mango juice	[94]
	Fruit concentrates and soils	[122]
	Various food and soil	[150]
PCR/RAPD-PCR	Orchard soil, soil on the fruit (pear, peach, apricot and apples) and samples of water and materials through the production line	[86]
	Soil from lemon orchard	[151]
	Soil of Foggia and pear juice	[152]
RAPD-PCR	Passion fruit juice	[95]
	N/A	[149]
PCR/ERIC-PCR	Orchard soil	[153]
PCR/PCR-DGGE	Fruit juices and fruit juice blends from Ghana and Nigeria	[80]
PCR/RT-PCR	Apple juice and saline	[154]
PCR-RFLP	Various juices and concentrates, drinks and intermediates	[155]
	Concentrated apple juice	[156]
	Concentrated apple juice and processing environment	[85]
	Orange juice	[157]
RT-PCR	Orange juice	[158,159]
	Orange juice, sports drink, lemonade and NaCl solution	[160]
	Apple juice	[161]
	Flavored non-carbonated drinks	[162]
	Acid buffer	[163]
IMS-PCR	Apple juice	[144]
	Sterile water, apple juice and kiwi juice	[164]
IMS-RT-PCR	Apple and kiwi fruit orchard and fruit juice production line	[165]
qPCR(quantitative)	Apple juice	[166]
16S rRNA sequencing	N/A	[60]
	Soil and water	[111]
	Various orchards	[76]
	Kiwi juice, fruit, soil and air of orchards and fresh cut and frozen fruit	[82]
**Analytical Methods**		
HPLC	Apple juice	[138,167,168]
	Flavored non-carbonated drinks	[162]
	Pear concentrate	[169]
	Fruit juices and fruit juice blends from Ghana and Nigeria	[80]
	Soil from lemon orchard	[151]
	Tomato puree	[170]
GC	Mixed fruit drink	[140]
GC-MS	Apple drinks, apple juice concentrate and orange juice	[171]
	Apple, orange and peach juice	[172]
	Fruit concentrate, flume water and vinegar flies	[173]
	Kiwi juice and fruit	[110]
	Apple juice	[174]
GC-MS/GC-Olfactometry	Orange juice	[131]
	Fruit concentrates and soils	[122]
GC-MS/SPME	Apple juice	[67,161,175,176,177,178]
	Apple, pear and orange juice	[179]
	Orange juice	[180,181]
Electronic Nose	Apple, pear and orange juice	[179]
	Flavored non-carbonated drinks	[162]
	Apple, orange and peach juice	[172]
	Apple and orange juice	[98]
	Apple juice	[178,182]
	Concentrated apple juice	[174]
	Mixed fruit juice beverage	[183]
	Orange juice	[181]
Fourier Transform Infrared Spectroscopy (FTIR)	Apple juice	[184,185,186]

N/A: Not Available.

**Table 3 foods-11-00747-t003:** Overview of chemical treatments applied for the inactivation of *Alicyclobacillus* species.

Chemical Treatments	Compounds	References
Oxidants	ClO_2_	[133,207,208,209]
	Ozone	[210]
	Sodium benzoate, Potassium sorbate	[103]
Natural compounds of microbial origin (bacteriocins)	Nisin	[89,129,217,218,219,220]
	Enterocin A5-48	[223]
	Bificin C6165	[224]
	Biovicin HC5	[225]
	Warnericin RB4	[226]
	Paracin C	[227,228]
	Cyclin A	[229]
Natural compounds of animal origin	Lysozyme	[234,235]
	Chitosan	[241]
Natural compounds of plant origin	Essential Oils	[90,244,245]
	Fatty acids and esters	[246,247]
	Plant extracts	[248,249,250]

**Table 4 foods-11-00747-t004:** Overview of HHP conditions applied for the inactivation of *Alicyclobacillus* species.

*Alicyclobacillus* Species	Medium	Experimental Conditions	Reference
*A. acidoterrestris* ATCC49025 *A. acidoterrestris* NFPA 1013 (apple juice isolate)	Apple juice	207, 414 and 621 MPa/1, 5 and 10 min/22, 45, 71 and 90 °C	[264]
*A. acidoterrestris* DSMZ 2492	BAM broth Orange juice Tomato juice Apple juice	Broth 350 and 450 MPa/5, 10 and 20 min/35, 45 and 50 °C Juices 350 MPa/20 min/50 °C and storage 3 weeks/30 °C	[265]
*A. acidoterrestris* DSMZ 2492	BAM broth	350 and 450 MPa/35, 45 and 50 °C	[266]
*A. acidoterrestris* NFPA 1101 (apple juice isolate) *A. acidoterrestris* NFPA 1013 (apple juice isolate)	Apple Juice concentrate	207, 414 and 621 MPa/1, 5 and 10 min/22, 45, 71 and 90 °C Various concentrations of juice (17.5, 35 and 70 °Brix)	[267]
*A. acidoterrestris* LMG 16906	Citric acid buffer (pH 4.0 and 5.0) Potassium phosphate buffer (pH 7.0) Tomato sauce (pH 4.2 and 5.0)	HHP and combined treatment HHP + heat Buffers 100, 200, 300, 400, 500 and 600 MPa/40 °C/10 min + heat 80 °C/10 min Tomato sauce 100, 200, 300, 400, 500 and 600 MPa/25, 40 and 60 °C/10 min + heat 80 °C/10 min	[268]
*A. acidoterrestris* TO-29/4/02 (apple juice isolate)	Apple juice	200, 300 and 500 MPa/30 min/50 °C continuously 100, 200, 300 and 500 MPa/2, 4 and 6 cycles of 5 min with 5 min pause/50 °C 100, 200 MPa × 6 cycles and 200 MPa x 4 cycles/5 min with 5 min pause/50 °C incubation 60 min at 50 °C/pressure 500 MPa/30 min 50 °C Combined treatment of HHP + lysozyme: 300 MPa/5, 10, 15 and 30 min/50 °C + 0.05 and 0.1 mg/mL lysozyme Combined treatment of HHP +nisin: ◦ 300 MPa/5, 10, 15 and 30 min/50 °C + 500, 750 and 1000 IU/mL nisin ◦ 200 MPa/5, 10, 15 and 20 min/50 °C + 250 IU/mL nisin	[237]
*A. acidoterrestris* NZRM 4098	Orange juice	200 and 600 MPa/1–15 min/45, 55 and 65 °C	[263]
*A. acidoterrestris* DSMZ 2498	Apple juice Orange juice	200, 400 and 600 MPa/10 min/20, 50 and 60 °C and storage for 28 days	[98]
*A. acidoterrestris* TO-29/4/02 (apple juice isolate) *A. acidoterrestris* TO-117/02 (apple juice isolate)	Apple juice (11.2, 23.6, 35.7 and 71.1 °Brix)	200 MPa/5, 10, 15, 30 and 45 min/50 °C 200 MPa/5, 10, 15 and 30 min/50 °C for 11 days/11 and 16 months spores of TO-29/4/02 200 MPa/5, 10, 15 and 30 min/50 °C for 10 days/2, 10, 11 and 23 months spores of 117/02 200 MPa/1, 5, 10, 15 and 30 min/50 °C × 3 subsequent treatments	[269]
*A. acidoterrestris* TO-117/02 (apple juice isolate)	Mcllvain buffer (pH 4.0 and 7.0) Apple juice	Germination and Inactivation 100, 200, 300, 400 and 500 MPa/20 min/50 °C (buffers and juice) 200 MPa/5, 10, 15 and 30 min/20, 50 and 70 °C (juice) 200 MPa/5, 10, 15 and 30 min/11.3, 23.7, 35.5 and 70.7 °Brix 200 and 500 MPa/20 min/50 and 70 °C (juice) 200 MPa/5, 10, 15 and 30 min/50 °C (buffers and juice) 200 MPa/50 °C for 2, 4 and 6 cycles/5 min with 5 min pause	[270]
*A. acidoterrestris* TO-117/02 (apple juice isolate)	Apple juice Buffer (pH 4.0)	200, 300, 400 and 500 MPa/15 min/4, 20 and 50 °C Determination with optical density	[271]
*A. acidoterrestris* NZRM 4447	Orange juice	200 and 600 MPa/15 min/39 °C + thermosonication 20.2 W/mL/78 °C	[127]
*A. acidoterrestris* NZRM 4447	Malt Extract Broth (10, 20 and 30 °Brix)	600 MPa/up to 45 min/35, 45, 55 and 65 °C 600 MPa/up to 45 min/45 °C Validation for apple juice, lime juice concentrate and Blackcurrant juice concentrate	[97]
*A. acidoterrestris* CCT 7547	Acai pulp	600 MPa/5, 10, 15, 20 and 25 min/65 °C	[272]
*A. acidoterrestris* CCT 7547	Deionized Water	300 and 600 MPa/5 min/25 and 70 °C + heat shock	[273]
*A. acidoterrestris* AJA 66 (apple juice isolate) *A. acidoterrestris* ATCC 49025	Apple juice Potassium Phosphate Buffer (pH 3.7 and 7.0)	600 MPa/1, 3 and 5 min/70, 80 and 90 °C	[274]
*A. acidoterrestris* (apple juice isolate) *A. acidoterrestris* DSMZ 2498	Orange juice	500 and 600 MPa/1, 3, 5, 15 and 30 min/25, 45, 60 and 70 °C	[275]
*A. acidoterrestris* CCT 7547	Mango pulp	600 MPa 0, 5, 10, 15, 20 and 25 min/65 °C 0, 4, 8, 12, 16 and 20 min/70 °C 0, 2, 4, 6, 8 and 10 min/75 °C 0, 1.5, 3, 4.5, 6 and 7.5 min/80 °C 0, 1, 2, 3, 4 and 5 min/90 °C	[276]
*Alicyclobacillus* spp. N1089 (canned tomatoes isolate) *Alicyclobacillus* spp. N1098 (apple juice isolate)	Tomato juice Apple juice	Combined treatment of HHP + Sucrose laurate L1695 392 MPa/10 min/45 °C + 0.005% and 0.01% for N1089 392 MPa/10 min/45 °C + 0.025, 0.04 and 0045% for N1098	[247]

## Data Availability

Not applicable.

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
