# Peer review of "Fruit Juice Spoilage by Alicyclobacillus: Detection and Control Methods—A Comprehensive Review"

_foods, 2022, doi:10.3390/foods11050747_

Round 1

Reviewer 1 Report

This is a well written review and I have only a few minor comments:

Heading to paragraph 3, line 272: I would add "...and associated spoilage." in the end as the paragraph also discuss methods for detecting spoilage.

l. 276: Perhaps use another word than "taint" - perhaps replace with "spoilage"?

l. 540: Perhaps just define what is meant by "super-dormant".

Table 3, first/second page: This species should be in top of table, not middle.

Author Response

We would like to thank the reviewer for the comments. All comments have been addressed appropriately in the manuscript.

Reviewer 2 Report

Please specify how the improvement of the esisting techniques and the development of new methods to ensure the rapid and early detection of Alicyclobacillus could be useful for the industry.  From an HACCP point of view, the controls carried on the raw materials and the implementation of the tecniques in order to minimize/eliminate the hazard are enough to preserve the quality of fruit juices. The methods actually available for the detection of Alicyclobacillus do You think are not enough to satisfy the necessity of sporadic controls performed by the Producers? 

Author Response

Response: We would like to thank the reviewer for the valuable comments. This comment has been covered in the first lines of paragraphs 3 and 4. We believe that the available methods employed by the industry are satisfactory to control the sporadic presence of Alicyclobacillus. However, a recent research survey undertaken by the authors, in orange and peach juices obtained from the market (unpublished data) resulted in the isolation of Alicyclobacillus acidoterrestris at 10% and 2% from orange and peach juices, respectively. The outcome of this survey indicates that new and improved methods of detection would be more useful for the industry for the effective control of the bacterium.

Reviewer 3 Report

The topic of your manuscript sounds interesting and an appropriate design. Overall information presented in this article provides a good foundation for future studies.

Need to improve the abstract according to the detection and control methods. Add some more information about these.

Author mentioned more than 300 references in this review article. I think it’s more than enough this kind of review article. I think author should remove the old references and should focused on 2011 to onward.

Authors need to explain the some mechanistic approaches of non-thermal techniques against the Alicyclobacillus. According to the title control methods should be described in detail. Because non-thermal techniques are very hot topic now a days.

If author add some graphical representation of data it better for readers.

Its better if author add some tabulated information of control methods such as physical treatment, plant origin, animal origin, microbial origin, and chemical treatments  

Author Response

We would like to thank the reviewer for the valuable comments.

Comment 1: The topic of your manuscript sounds interesting and an appropriate design. Overall information presented in this article provides a good foundation for future studies.

Response: Thank you very much for the comments.

Comment 2: Need to improve the abstract according to the detection and control methods. Add some more information about these.

Response: The abstract has been amended as suggested and the requested information has been added. Please see revised abstract.

Comment 3: Author mentioned more than 300 references in this review article. I think it’s more than enough this kind of review article. I think author should remove the old references and should focus on 2011 to onward. Response: An effort of reducing the references has been undertaken as recommended.

Comment 4: Authors need to explain the some mechanistic approaches of non-thermal techniques against the Alicyclobacillus. According to the title control methods should be described in detail. Because non-thermal techniques are very hot topic nowadays.

Response: We would like to thank the reviewer for this comment. In the revised version of our manuscript, we have included the requested information on the mode of action of non-thermal treatments to inactivate Alicyclobacillus. Please see revised paragraph 4.3 on page 14.  

 Comment 5: If author add some graphical representation of data it better for readers. Its better if author add some tabulated information of control methods such as physical treatment, plant origin, animal origin, microbial origin, and chemical treatments

Response:  Following the suggestion of the reviewer a new table has been included in the revised manuscript as Table 3, including tabulated information on the chemical treatments applied for the control of Alicyclobacillus species.

Round 2

Reviewer 3 Report

Author revised the article accordingly.